# Effects of Backfilling Excavated Underground Space on Reducing Acid Mine Drainage in an Abandoned Mine

**Kohei Yamaguchi [1,2,*], Shingo Tomiyama [3], Toshifumi Igarashi [3], Saburo Yamagata [4], Masanori Ebato [5] and Masatoshi Sakoda [6]**

[1] Division of Sustainable Resources Engineering, Graduate School of Engineering, Hokkaido University, Sapporo 060-8628, Japan

[2] Mitsubishi Materials Corporation, 1-600, Kitabukuro-cho, Omiya-ku, Saitama-shi, Saitama 330-8508, Japan

[3] Faculty of Engineering, Hokkaido University, Sapporo 060-8628, Japan; tomiyama@mmc.co.jp (S.T.); tosifumi@eng.hokudai.ac.jp (T.I.)

[4] Mitsubishi Materials Corporation, 3-2-3, Marunouchi, Chiyoda-ku, Tokyo 100-8117, Japan; s-yamaga@mmc.co.jp

[5] Oyo Corporation, 2-10-9, Daitakubo, Minami-ku, Saitama-shi, Saitama 336-0015, Japan; ebato-masanori@oyonet.oyo.co.jp

[6] Japan Oil, Gas and Metals National Corporation (JOGMEC), 2-10-1, Toranomon, Minato-ku, Tokyo 105-0001, Japan; sakoda-masatoshi@jogmec.go.jp

**\*** Correspondence: kyama@mmc.co.jp; Tel.: +81-48-641-5691

**Abstract:** Three-dimensional groundwater flow around an abandoned mine was simulated to evaluate the effects of backfilling the excavated underground space of the mine on reducing the acid mine drainage (AMD). The conceptual model of the groundwater flow consists of not only variable geological formations but also vertical shafts, horizontal drifts, and the other excavated underground space. The steady-state groundwater flow in both days with high and little rainfall was calculated to calibrate the model. The calculated groundwater levels and flow rate of the AMD agreed with the measured ones by calibrating the hydraulic conductivity of the host rock, which was sensitive to groundwater flow in the mine. This validated model was applied to predict the flow rate of the AMD when backfilling the excavated underground space. The results showed that the flow rate of the AMD decreased by 5% to 30%. This indicates that backfilling the excavated space is one of the effective methods to reduce AMD of abandoned mines.

**Keywords:** abandoned mine; groundwater flow analysis; acid mine drainage; backfilling

## 1. Introduction

Acid mine drainage (AMD) is generated at many active, closed, and abandoned mines throughout the world. AMD is a serious environmental issue in the mining industry [1–9], which is generally characterized by low pH and high concentrations of sulfate, heavy metals (e.g., copper (Cu), lead (Pb), zinc (Zn), and cadmium (Cd)) [10,11], and metalloids (e.g., arsenic (As), [12]). Low pH is caused by oxidation of sulfide minerals and dissolves heavy metals in the host rock. Moreover, the contribution of bacteria is important under in situ conditions for AMD formation [13,14]. The chemical reactions (1)–(4) [15] causes low pH when the oxygenated rainwater contacts with pyrite in the unsaturated zone. Backfilling the excavated underground space may reduce the amount of AMD due to the decrease in hydraulic gradient [16,17]. In addition, the contact area between oxygenated rainwater and pyrite decreases due to the rise of groundwater levels.

$$FeS_2(s) + 7/2O_2 + H_2O = Fe^{2+} + 2SO_4^{2-} + 2H^+ \tag{1}$$

$$Fe^{2+} + 1/4O_2 + H^+ = Fe^{3+} + 1/2H_2O \tag{2}$$

$$Fe^{3+} + 3H_2O = Fe(OH)_3(s) + 3H^+ \tag{3}$$

$$FeS_2(s) + 14Fe^{3+} + 8H_2O = 15Fe^{2+} + 2SO_4^{2-} + 16H^+ \tag{4}$$

Sulfide minerals in the host rock exposed to shallow groundwater or rainwater are oxidized during the operation of the mine. In addition, AMD is continuously generated for more than several decades after closing or abandoning the mines [18].

In Japan, there are over 5500 closed or abandoned non-ferrous metal mines and 79 mines of these are treating AMD. AMD from closed or abandoned mines is commonly treated by neutralization with hydrated lime or sodium hydroxide. In this application, most toxic heavy metals are precipitated and removed [19], and then the treated water is released into nearby rivers. The treatment of AMD induces a large load from an economic perspective although lime neutralization has effectively been used over the last 40 years in Japan. Total subsidiary aid cost for about 50 years is ~70 billion JPY (~650 million USD) in Japan [20].

Active treatments by adding alkaline reagents are costly and should last for decades. On the other hand, passive treatments are expected to be applied to mines with a relatively lower flow rate of AMD [21]. For both treatments, it is necessary to reduce the amount of produced AMD. Thus, it is important to clarify the processes of generating AMD. There have been a variety of studies of AMD monitoring and characterization in not only Japan (e.g., [22–24]) but also other countries [25–34]. Some studies pointed out that groundwater flow patterns in and around mines should be evaluated to purpose countermeasures against AMD reduction. There are several methods of the mitigation, such as covering of ground surface with low-permeable layers and impoundment of land subsidence [35–41]. In this study, the effects of backfilling the underground space in a mine on produced AMD were examined by 3-D groundwater flow model to reduce the AMD produced because the mine selected has a huge volume of underground space already excavated.

## 2. Geology and Mining of Study Area

The selected mining area is located in valley terrain at an altitude of 300 to 400 m (Figure 1). The basement rock mostly consists of pre-Neogene granite. Neogene tuff, andesite, and rhyolite were deposited on the granite basement. They erupted and deposited on the granite in the marine. The type of mine is a vein-type deposit formed in faults and fractures in the Neogene strata. Distribution of the mined area was recorded in the pit map created during the operating period when drifts were excavated at a depth of 60, 90, and 150 m from the ground surface (−2 L, −3 L, and −5 L levels, respectively; Figure 1). Among them, the −5 L level drift is used as drainage of mine water to the mine mouth. Mining was carried out by the shrinkage method at the beginning of the operation because the host rock of the Neogene strata was solid and because the veins in the Neogene strata were inclined with slopes of 70° to 90°. Since the mine is producing sludge generated from AMD neutralization, backfilling the formed sludge mixed with cement (sandy slime) into excavated underground space was adopted to prevent collapses of the space. This is because the host rock to be excavated became crumblier with the progress of operation.

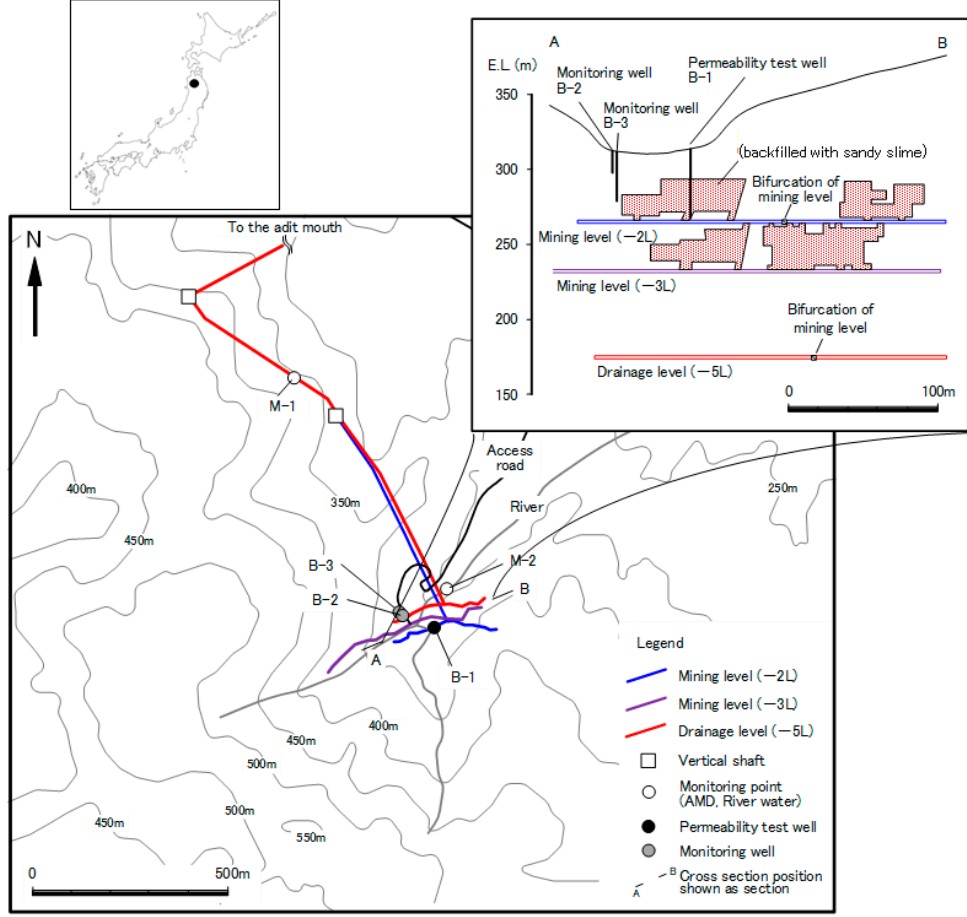

**Figure 1.** Distribution of old mining levels and excavated underground space of the study area.

## 3. Conceptual Model of Groundwater Flow

Groundwater flows through aquifers consisting of surface soil and weathered layers at this mine site. Groundwater in the aquifers flows to mining areas through faults and veins as shown in Figure 2 [42]. Groundwater in the mined area flows through the mining levels (−2 L, −3 L, and −5 L) to the adit mouth of the drainage level (−5 L). Thus, the total head of groundwater decreases from the ground surface to the deeper underground, and thereby duplicate aquifers, a shallower aquifer, and a deeper aquifer, are formed.

The flow rate of AMD is about 4 to 10 m³/min from this mine. The AMD from the adit mouth of −5 L level accounts for about 0.04 to 0.16 m³/min.

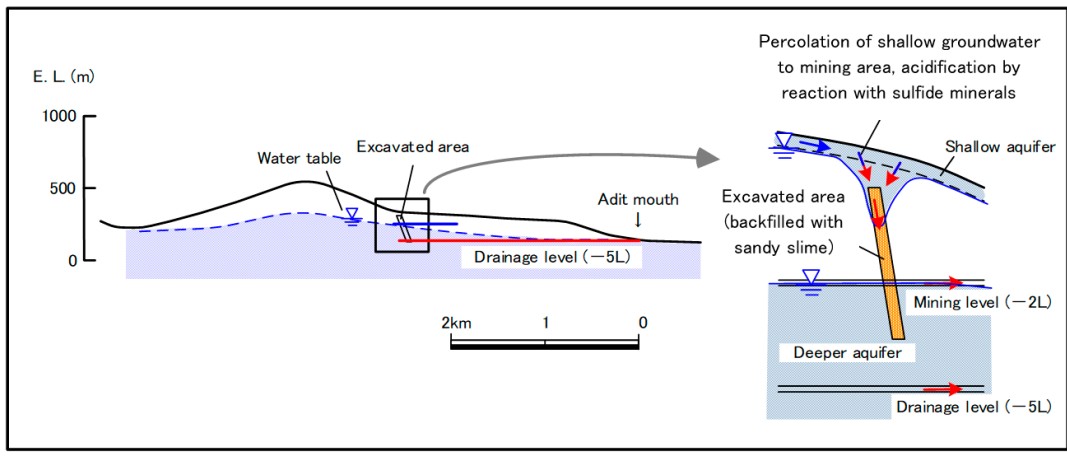

**Figure 2.** Conceptual model of groundwater flow and its discharge.

## 4. Methods

### 4.1. In Situ Survey

Groundwater levels were continuously measured at B-2 and B-3 from August 4, 2014 to December 17, 2016 at intervals of 60 min. The strainer pipes were installed from 15.0 to 35.0 m deep of B-2 well and from 3.0 to 15.0 m deep of B-3 well. B-2 is located less than 10 m away from B-3 and both wells are at the same ground level. This means that B-2 is to monitor the deeper groundwater level whereas B-3 is to monitor the shallower groundwater level. The location of these boreholes corresponds to upstream of groundwater into underground space, sensitive to the AMD production.

The flow rate of AMD was measured at M1 in Figure 1 using triangular weirs installed in the drain of the tunnel. The flow rate of AMD was measured from 4 August 2014 to 17 December 2016 at intervals of 60 min. The daily precipitation was calculated by accumulating hourly precipitations measured at the observatory 2 km away from the study area.

### 4.2. Theoretical Equation

Saturated–unsaturated groundwater flow analysis was applied to evaluate groundwater flow using Dtransu-3D-EL software [42,43] together with G-TRAN/3D pre- and post-processing softwares for Dtransu-3D [36,40]. Dtransu-3D-EL software solves the equation for saturated–unsaturated groundwater flow derived from the mass preservation and Darcy's equation, which can be written as

$$\rho_f \theta \gamma \frac{\partial c}{\partial t} + \rho \{\beta Ss + Cs(\theta)\} \frac{\partial \varphi}{\partial t} = \frac{\partial}{\partial x_i} \{\rho K_{ij}^S K_r(\theta) \frac{\partial \varphi}{\partial x_j} + \rho K_{i3}^S K_r(\theta) \rho_r\} \tag{5}$$

where $\phi$ is the pressure head, $\theta$ is the (volumetric) water content, $Ss$ is the specific storage, $Cs(\theta)$ is the specific water capacity, $K^s_{ij}$ is the directional components of the saturated hydraulic conductivity function, $K_r$ is the relative hydraulic conductivity, $t$ is time, $\rho_f$ is the density of solvent, $\varrho$ is the density of fluid, $\rho_r$ is the ratio of $\rho_f$ to $\rho$, $\beta = 1$ is the saturated zone, $\beta = 0$ is the unsaturated zone, and $\gamma$ is the solute density ratio [43].

### 4.3. Numerical Model

Basic configuration of the numerical model is shown in Table 1. The model domain had an area of 0.58 km$^2$ with a total elevation of 520 m and bounded by topographic ridges as shown in Figure 3. The mined area above the mining levels (−2 L, −3 L, and −5 L levels) was assumed to be in an unsaturated zone. The ground surface topography was reproduced by using the digital elevation model (DEM) created based on numerical maps by aerial survey. Ore body, mining levels (−2 L, −3 L, and −5 L) and excavated area (backfilled with sandy slime) were constructed in the mesh diagram of the numerical model based on the handwritten ore map and level map with 1:3000 scale. First, the ground plane was divided into squares with a side of 20 m. Next, a two-dimensional square on the ground surface were extended in the underground direction to form quadrangular prisms. Quadrangular prisms were added downward to create a three-dimensional mesh model.

**Table 1.** Basic configuration of the numerical model.

| Items | | Configurations |
|---|---|---|
| | Side and bottom | Impermeable |
| | Ground surface | Infiltration: 5.0 mm/day (days with high rainfall) 0.47 mm/day (days with little rainfall) |
| Boundary conditions | River | Pressure head: 0 m |
| | Mining levels (−2 L, −3 L, and −5 L) | Seepage |

| Finite element method grid | Number of elements | 19,890 |
| --- | --- | --- |
| | Number of layers | 12 |
| | Basic element size | x = 20 m; y = 0.5–140 m; z = 20 m |
| Method of analysis | | Saturated-unsaturated three-dimensional seepage analysis |

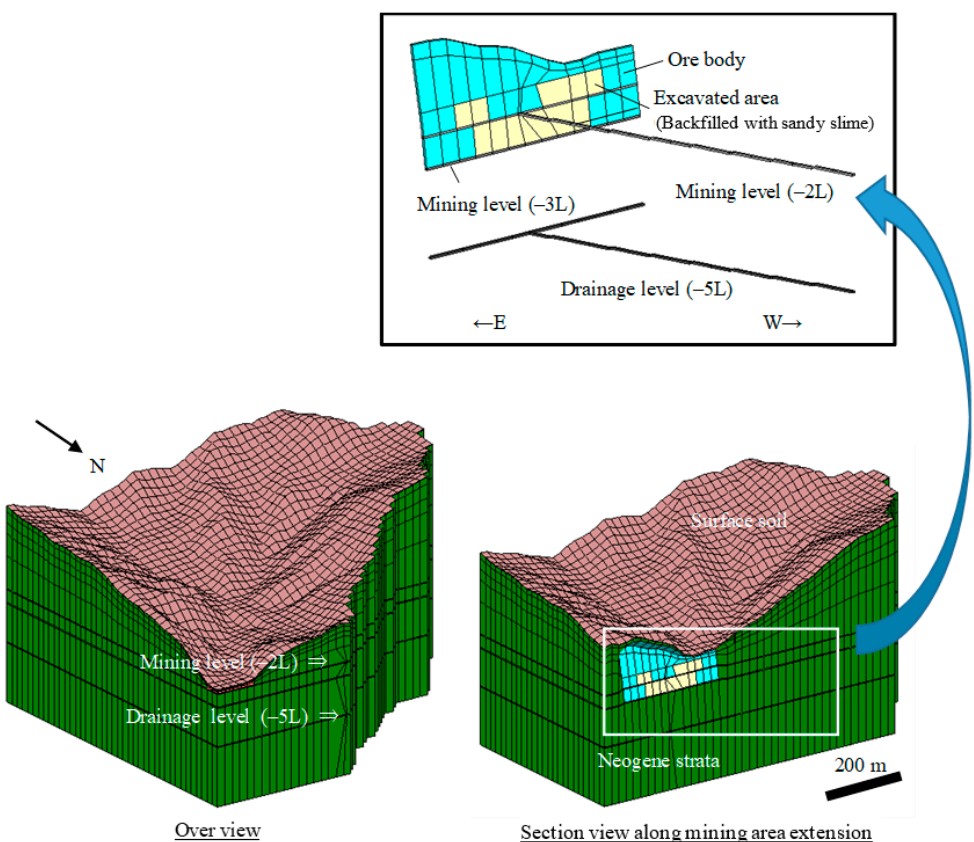

**Figure 3.** Simulation model of three-dimensional groundwater flow.

The hydraulic conductivities (*K*) and unsaturated properties of numerical blocks representing the surface soil, embankment, and Neogene strata as well as the excavated areas were based on either in situ measured results or estimated from the results of other papers as listed in Table 2. Hydraulic conductivities were obtained from in situ permeability tests or estimated from rock permeability data collected throughout Japan [44].

The properties of various geological strata under unsaturated conditions were obtained by the van Genuchten model [45], which are given by

$$S_e(\varphi) = \frac{\theta - \theta_r}{\theta_s - \theta_r} = [\frac{1}{1 + |a\varphi|^n}]^m \tag{6}$$

$$K_r\{S_e(\varphi)\} = S_e^l[1 - (1 - S_e^{1/m})^m]^2 \tag{7}$$

where $S_e$ is effective water saturation, $\theta_s$ is the saturated water content, $\theta_r$ is the residual water content, *a*, *m*, and *n* are the van Genuchten parameters [45], and *l* is a parameter representing the degree of pore connectivity (no unit). The van Genuchten parameters used for the current simulations are presented in Table 3, and the corresponding functions are depicted in Figure 4.

**Table 2.** Hydraulic conductivities of different geological strata.

| Type of Geological Stratum | Hydraulic Conductivity $K$ (m/s) | Reference |
|---|---|---|
| Surface soil | $1.1 \times 10^{-5}$ | [44] |
| Embankment | $7.0 \times 10^{-7}$ | [46] |
| Neogene strata | $1.0 \times 10^{-9}$, $9.2 \times 10^{-9}$, $1.9 \times 10^{-8}$, $9.4 \times 10^{-8}$ | in situ measurements |
| Ore body | $1.9 \times 10^{-6}$ | in situ measurements |
| Mining levels (−2 L, −3 L, and −5 L) | $1.0 \times 10^{-1}$ | [40] |
| Excavated area (backfilled with sandy slime) | $7.3 \times 10^{-6}$ | in situ measurements |

**Table 3.** Residual water content θ r, saturated water content θ s, and van Genuchten parameters *a*, *l*, and *n* [45].

| | θ r (-) | θ s (-) | $a$ (m$^{-1}$) | $l$ (-) | $n$ (-) | Reference |
|---|---|---|---|---|---|---|
| Tailings | 0.00 | 0.42 | 0.15 | 0.5 | 1.87 | [46] |
| Sand slime | 0.01 | 0.29 | 3 | 0.5 | 3.72 | [47] |
| Andesite | 0.00 | 0.104 | 0.06 | 0.5 | 3.57 | [48] |

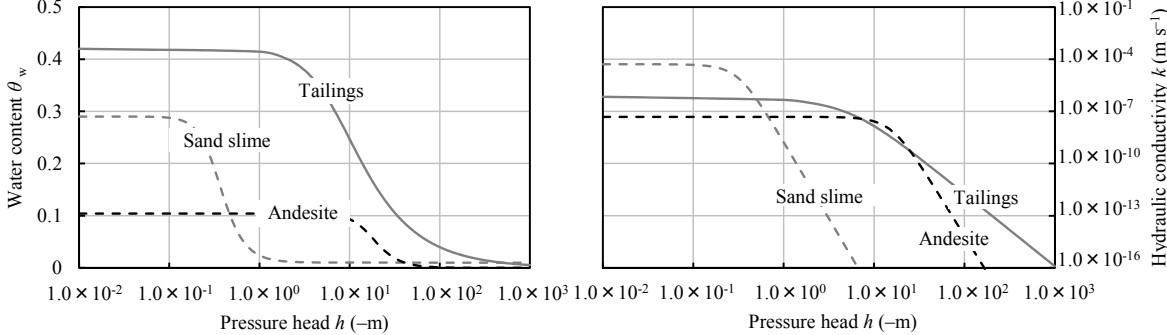

**Figure 4.** Water content and hydraulic conductivity curves using the van Genuchten's model for sandy slime, tailings, and andesite. See Table 3 for the corresponding parameters of the van Genuchten model [45].

The distribution of mined areas and tunnels (−2 L, −3 L, and −5 L levels) in the numerical model was based on the records of the operating period. At that time, the thickness of the mining area was set at 3 m based on the actual space size. Similarly, the height and width of mining levels were set at 2 m. The average rainfall in August 2014 was 13.5 mm/day (hereafter, days with high rainfall) and the average in August 2015 was 3.5 mm/day (hereafter, days with little rainfall). The infiltration rate was calculated based on the water balance analysis, days with high rainfall: 5.0 mm/day (recharge rate 0.37) and days with little rainfall: 0.47 mm/day), (recharge rate 0.13). Both mining level (−2 L level) and drainage level (−5 L level) were assumed to be seepage boundary conditions.

The boundary conditions of the numerical simulations are as follows:

- The river was assumed to be connected to the groundwater surface and was set as a fixed head boundary condition (pressure head 0 m).
- The ground surface (except that of the river) was set as infiltration boundary condition (Table 1).

The flow chart of the simulation is the same as Nishigaki (1995) [49].

## 5. Results and Discussion

### 5.1. Monitored Data

The groundwater level of B-2 in the deeper aquifer was almost constant at the elevation of about 287 m while that of the B-3 in the shallow aquifer was changed in response to rainfall. This means that the groundwater level of B-3 is sensitive to rain whereas that of B-2 is not so sensitive to rain.

The AMD flow rate is shown in Figure 5. The maximum value was 0.16 m³/min in April during the snowmelt season. Although the AMD flow rate tended to increase during the snowmelt season, the AMD during the summer varied from year to year, high in 2014 and low in 2015. This means that the infiltration of rainwater directly affects the flow rate of the AMD.

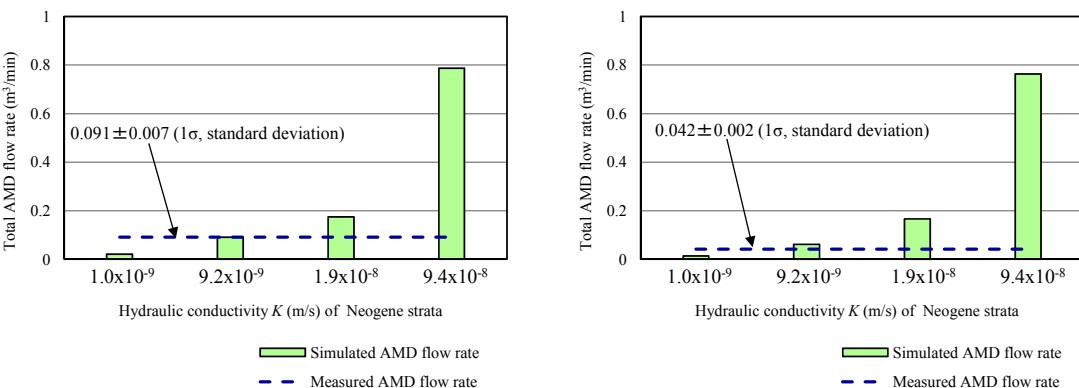

**Figure 5.** Comparison between the simulated and measured acid mine drainage (AMD) flow rate of days with high (**right**) and little (**left**) rainfall.

### 5.2. Calibration of Hydraulic Conductivity of the Neogene Strata

The study area is snowy in winter and snowmelt increases the amount of AMD in spring (March to May). However, the rain depends on the year during summer. Thus, the AMD flow rates in days with high rainfall (August 2014) and days with little rainfall (August 2015) were selected for calibration. In order to evaluate seasonal variations of AMD flow rates, analysis of steady state groundwater flow was performed for both days with high and little rainfall.

The simulated AMD flow rates by changing the hydraulic conductivity of the Neogene strata ranging from $1.0 \times 10^{-9}$ m/s to $9.4 \times 10^{-8}$ m/s are compared with those measured rates as shown in Figures 5 and 6. Since measured hydraulic conductivity of the Neogene strata ranged from $1.0 \times 10^{-9}$ m/s to $9.4 \times 10^{-8}$ m/s, the hydraulic conductivity was parametrically changed in the simulation. The calculated AMD flow rate strongly depended on the hydraulic conductivity of the Neogene strata. The AMD flow rate was the highest in case of the highest hydraulic conductivity of $9.4 \times 10^{-8}$ m/s, and lowest in case of the lowest hydraulic conductivity of $1.0 \times 10^{-9}$ m/s. When the hydraulic conductivity of the Neogene strata was $9.2 \times 10^{-9}$ m/s, the flow rate of AMD agreed with the measured ones during both days with high and little rainfall. This means that the same hydraulic conductivity of the Neogene strata is applicable to any season.

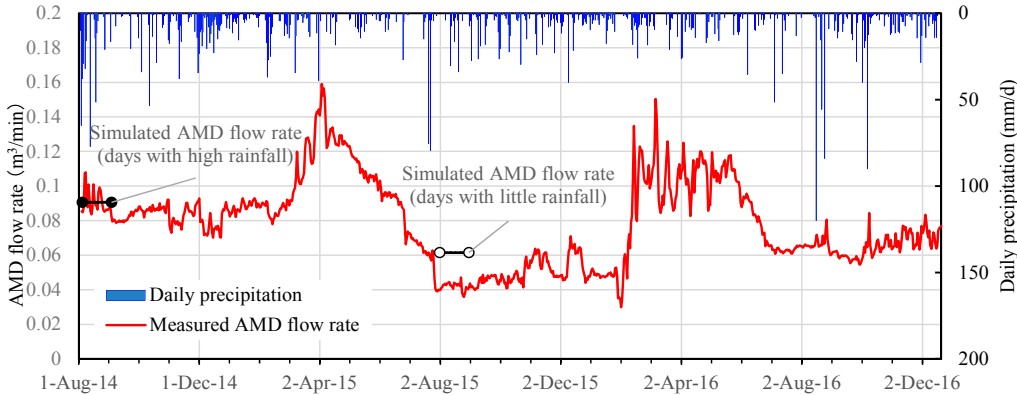

**Figure 6.** Change of precipitation and AMD flow rate.

The sensitivity analysis of hydraulic conductivity of the ore body was also conducted to investigate whether the ore body distributed on the upper part of −5 L affected the flow rate of AMD. When the hydraulic conductivity of the ore body was reduced to $1.9 \times 10^{-5}$ m/s, the AMD increased by 20%, and when the hydraulic conductivity was increased to $1.3 \times 10^{-5}$ m/s, the AMD decreased by 10%. This indicates that the effect of hydraulic conductivity of the Neogene strata on AMD flow rate was more significant than that of the ore body. It was also found that the amount of AMD from the deeper −5 L decreased when the shallower ore body collected more groundwater.

Calculated groundwater levels are compared with observed ones in boreholes as shown in Figure 7. The measured groundwater levels agreed with the calculated ones, irrespective of seasons when the hydraulic conductivity of the Neogene strata of $9.2 \times 10^{-9}$ m/s was used. The measured and simulated levels of B-3 increased during the days with high rainfall and decreased during the little rain days. The simulated groundwater level of B-2 had a difference of about 4 m between the days with high and little rainfall. However, the measured values of B-2 were constant during both seasons. This indicates that the groundwater flow model can simulate the shallower groundwater flow and not deeper groundwater flow well.

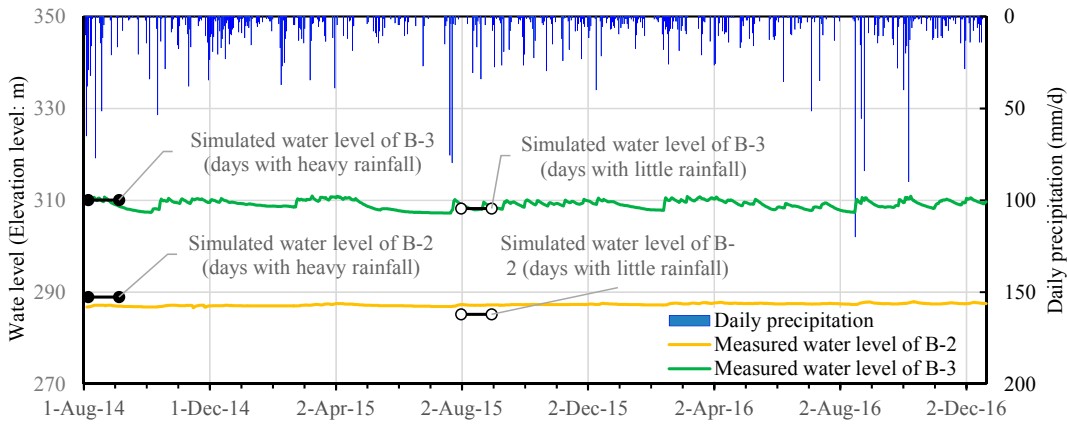

**Figure 7.** Change of precipitation and groundwater levels at B-2 and B-3.

Figure 7 shows changes in groundwater levels of monitoring wells of B-2 and B-3. The simulated results agreed with the observed ones. This implies that the calibrated model is effective in evaluating groundwater levels of both rainwater-sensitive and rainwater-insensitive wells.

The simulated value of B-2 responds to the increase/decrease in rainfall with a water level difference of 3.8 m between the days with high rainfall and little rainfall. Even if the amount of rainfall is large, if it runs off the surface layer, it may not contribute to the rise in water level. Therefore, we conducted a sensitivity analysis in which the hydraulic conductivity of the surface soil was increased by 10%. As a result, the water level difference between days with high rainfall

and little rainfall was 3.7 m. Although it had the effect of suppressing the increase/decrease in water level difference, it was small.

*5.3. Prediction of the Effect of Backfilling*

Figure 8 shows the vertical distribution of total head and Darcy velocity along the drainage tunnel simulated by 3-D groundwater flow model. Before backfilling the underground space, the total head decreased from the ground surface to the mining level (−2 L level) as shown in Figure 8b. Both groundwater surfaces of the shallower and lower levels around the mining level (−2 L level) were clearly observed in the simulation (Figure 8b). The AMD flow rate exceeded 1.0 m/day as Darcy flow velocity around the mining level (−2 L) (Figure 8c). This flow rate corresponds to 0.091 m³/min of AMD produced. After backfilling the mining levels (−2 L, −3 L, and −5 L), AMD flow rate was calculated at 0.059 m³/min, 64.9% of AMD flow rate before backfilling, during days with high rainfall (35.1% reduction). In days with little rainfall, AMD flow rate was calculated at 0.059 m³/min after backfilling, which was 95.1% of the flow rate before backfilling (4.9% reduction). This indicates that the AMD flow rate was reduced after backfilling whether it was the rainy season or not. In particular, backfilling the underground space was more effective during the rainy season. On the other hand, the groundwater level rose after the mined area was backfilled (Figure 8b). In addition, the groundwater flow around the −2 L level was reduced after the backfilling (Figure 8c). Higher groundwater level could mitigate the reaction between atmospheric oxygen and sulfide minerals in the underground. Thus, the rise in groundwater level may prevent AMD formation. If DO (Dissolved Oxygen) is consumed in the saturated zone, there is no new supply of oxygen and the oxidation reaction of pyrite does not occur. The geochemical modelling like previous theoretical model and experiments [50–52] is necessary to estimate the amount of acidification.

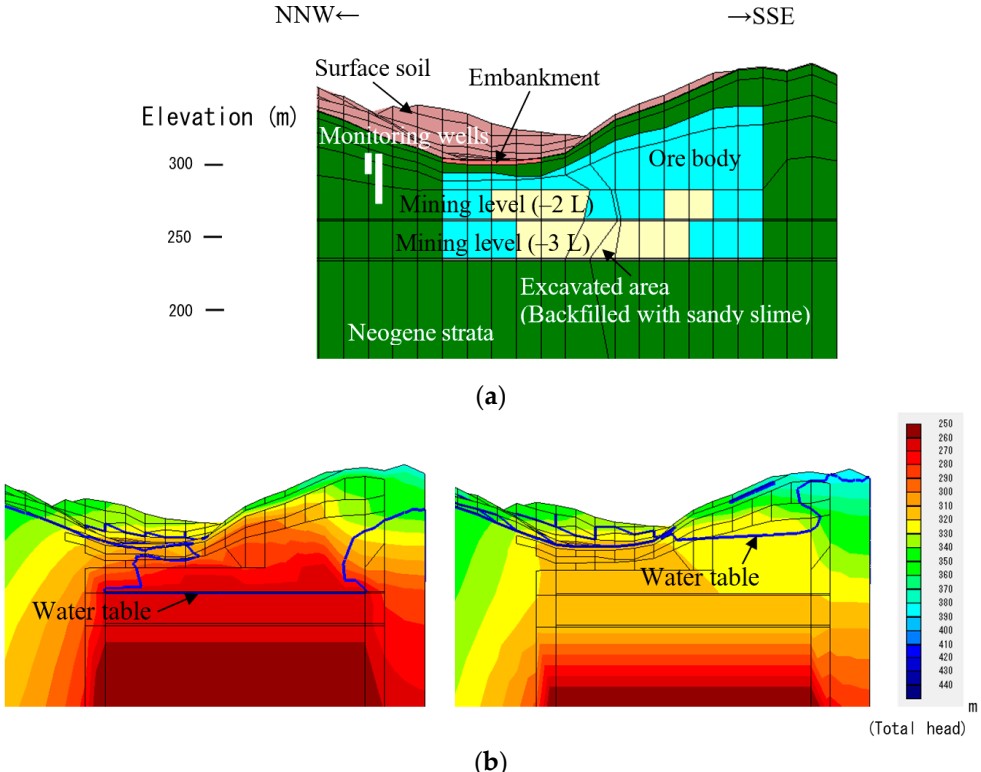

(**a**)

(**b**)

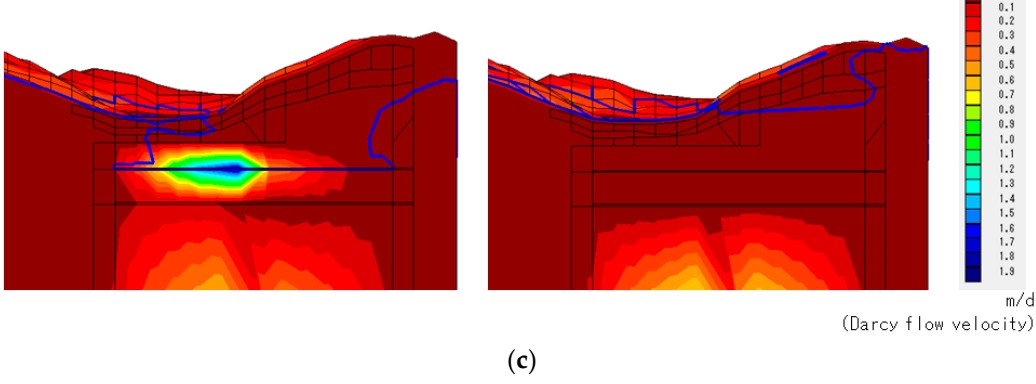

(**c**)

**Figure 8.** Vertical section of simulation model and simulated results of groundwater flow. (**a**) Vertical section of simulation model. (**b**) Distribution of total heads before (left) and after backfilling (right). (**c**) Distribution of Darcy flow velocity before (left) and after backfilling (right).

The measured amounts of AMD showed a difference of about twice in the days with high rainfall and the days with little rainfall as shown in Figure 8 and 1.5 times in the simulation. The amount of AMD in the days with high rainfall and that in the days with little rainfall were almost the same after backfilling because AMD was not significantly affected by the infiltration of rainwater.

The obtained results imply that backfilling underground space is effective in reducing the flow rate of AMD and preventing AMD formation is also expected. The backfilling method is promising when inclined water-conductive zones, such as excavated underground space and fractures, are located in the mine. Based on the outcome of this study, an appropriate design of backfilling underground space can be performed to mitigate the AMD in closed and abandoned mines. The effects of backfilling underground space on AMD production depend on the geology and hydrogeology of mine sites, so applying this method to a variety of sites is required to quantitatively evaluate the performance.

## 6. Conclusions

3-D numerical model consisting of a variety of geological formations and underground tunnels was constructed. By using the calibrated model, both groundwater levels around the mine and flow rates of AMD agreed with the measured values, irrespective of the season. In this mine, AMD after backfilling the underground space was reduced to more than 30% in days with high rainfall and to 5% in days with little rainfall. In addition, the acidification of groundwater may be expected due to the rise in groundwater levels. These results imply that backfilling the underground space is effective in reducing AMD in this mine.

**Author Contributions:** Conceptualization, K.Y. and S.T.; software, K.Y. and S.T.; validation, K.Y., S.T. and S.Y.; investigation, M.E.; writing—original draft preparation, K.Y. and S.T.; writing—review and editing, T.I.; visualization, K.Y. and S.T.; supervision, M.S. All authors have read and agreed to the published version of the manuscript.

**Funding:** This work was supported in part by Development of advanced technology for mine drainage treatment from 2012 to 2014 of Ministry of Economy, Trade, and Industry.

**Acknowledgments:** The authors wish to thank the editor and anonymous reviewers for their constructive comments that improved this manuscript. We would also like to thank the staffs of Mitsubishi Materials Corporation, Eco-Management Corporation, and Mitsubishi Materials Techno Corporation for their help, advice, and cooperation during this study.

**Conflicts of Interest**: The authors declare no conflict of interest. The funders had no role in the design of the study; in the collection, analyses, or interpretation of data; in the writing of the manuscript, or in the decision to publish the results.

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
