# Peer review of "Effects of Backfilling Excavated Underground Space on Reducing Acid Mine Drainage in an Abandoned Mine"

_minerals, doi:10.3390/min10090777_

Round 1

Reviewer 1 Report

Dear Author,

This is my review of the paper untitled “Effects of backfilling excavated underground space on reducing acid mine drainage in an abandoned mine”, by K.Yamaguchi and co-author. To me the study is promising (the model is interesting by its formulation, its main results and its opportunity to be adapted to other studies) but not complete at this level. Indeed, the way the proposed numerical model works has to be better described to put it forward, the results also have to be developed, and finally, the more important to me, a true discussion is needed. I insiste on the fact that the discussion needs to be separated from the results. Indeed, if you give your interpretations and discussion in the same time as your results, you induced a bias in the reader mind. We have to be able to read your results whithout being influenced by your thought in order to make our own ones and confront them to yours. I develop these three points below.

1. the numerical model

it is not clear how it works. two information would be useful to help the reader to understand how and if they can adapt the study to their case: how the geometry is input and a general flowchart of the algorithm. Also the mesh generation could be better described together with the theoretical context (or more references could be given here)

2. the results

the results are so frustrating. they are not enough described, quantified, criticized. The errors between the prediction and the measured value have to be measured and discussed (in the discussion section).

also Fig7 need to be developped or deleted (two sentences will resume it at this stage)

3. the discussion. it is not because you write "results and discussion" that you really provide a discussion. and here, there is no discussion. Discussion is important. It is even more, it is crucial, it gives the opportunity to describe the uncertaincy, to give personnal interpretations, to open your study and finally to interest the reader. I see at least four points that could be developed :
  • the interpretation of the results and the conclusion about the validation (or not) of the model.
  • The error discussion (for example in Fig8, the model’s results are perfectly following B3 water level. But they are not to B2 water level, is this difference acceptable ? why ? how to improve it ? same question for fig 6 and the second simulated amd flow versus the measured one).
  • The sensibility of the model to different physical and/or numerical parameter (what happen when the fluid viscocity or other is changed from 5%, 10% etc ? what is the stability of the results when changing the mesh sizes, or the number of increments etc?...)
  • And finally (but certainly not the least point) the paper could be open to different application (transfert of underground pollution for example) or to different scale (laboratory investigations). Indeed, the paper is too much self-focused, so the readers could not see their interest in it and the paper could become useless.

About this last point, the introduction and general bibliography is also too much self-focused (in terms of scale, application and topic).

Sincerely yours

Author Response

Dear reviewer,

Thank you veriy much for your review.

Best regards,

Kohei Yamaguchi

Reviewer 2 Report

Journal: Minerals (ISSN 2075-163X)

Manuscript ID: minerals-850930

Title: Effects of backfilling excavated underground space on reducing acid mine drainage in an abandoned mine

Reviewer’s Comments

The manuscript presents data from a conceptual three-dimensional groundwater flow model simulation around an abandoned mine in Japan to evaluate the effects of backfilling of the excavated underground space of the mine on reducing the acid mine drainage (AMD). The model showed that after backfilling the underground space the flow rate of the AMD was decreased by 5% (in little rain days) to 30% (in heavy rain days).

The manuscript is missing few important information and justification and can only be accepted after the concerns are addressed properly.

  1. The introduction should include the mechanism behind AMD formation in an abandoned mine. How Pyrite after being exposed to air and water reacts and forms AMD should be clearly stated for the general readers. The manuscript is introducing a technique that will inhibit/slow down this chemical reaction. Hence, this information should be added and the chemistry behind this proposed backfilling technique should be explained properly.
  2. What other commonly used AMD control technologies exist (other than the lime addition) should be mentioned briefly in the introduction section. What are the advantages of this backfilling technique over the other techniques should also be mentioned here.
  3. Thoroughly revise the manuscript for the spelling errors such as in Line 62 it should be “space” instead of “apace.” A thorough revision for correcting the grammatical errors is also needed.
  4. In Figure 2, one of the text boxes got cut. Please edit it properly.
  5. What is the amount of AMD already present in the underground mine site? The model is mentioning about reducing the AMD in the underground space but did not provide the extent of the existing AMD pool. This is a very important information for this type of AMD management.
  6. The manuscript talked about two boreholes (B-2 and B-3). It mentioned that “the groundwater level of B-3 was sensitive to rain whereas that of B-2 was insensitive to rain.” But no details have been provided on the location of these boreholes. Any inference about the borehole related data would not be appropriate without knowing their exact location.
  7. The manuscript referred “little rain days” and “heavy rain days” without providing rainfall amount. This information along with average rainfall amount of the study area should be included.
  8. Figure 7 do not have any standard deviation (SD). Without SD this graph could not be scientifically accepted.
  9. The manuscript many times mentioned “…improve the quality of the AMD.” This phrase is not scientifically correct, as you cannot improve the quality of the AMD, either you can reduce the AMD generation, or you can treat it (using the treatment technologies). So, please change this phrase throughout the manuscript with more appropriate term.
  10. More discussion is needed by correlating the simulated and monitored data. In its current form, the “Results and Discussion” section is only showing the results without adequately discussing the logic behind them. This section should be rewritten by properly explaining what the obtained results are implying.
  11. The manuscript is written based on only one set of data point. Explanation should be included to justify its repeatability (multiple little and heavy rain data points and their correlation).
  12. The “Methods” section elaborately described the simulation part, but it failed to provide the specific details of the measured field data collection part. This detail should be provided.

Author Response

Dear reviewer,

Thank you very much for your review.

Please fine an attachment.

Best regards,

Kohei Yamaguchi

Round 2

Reviewer 1 Report

Dear author,

Apart an improvement of the method description (that is still incomplete), I do not consider that my previous remarks have been answered. I'm surprise by the rapidity of your return and I think you misunderstood the principle of a major review. It needs a true supplementary work, not just a couple of added sentences. Bibliography need to be more investigated/analysed, your paper is (as previously) too much self focused. the results are too brief and too briefly described. there is no discussion: no critics of your results, no sensibility studied, no oppening/comparison to other application, other scale, other bibliography... ... ...

Finally about your answer byitself. A major review need more than an answer based on "I added line x to y" without any exaplnation, details, added informations etc.

Author Response

Thank you for your review.

I upload my responce.

Reviewer 2 Report

The authors have made significant changes to improve the quality of the manuscript. All the issues raised previously have been addressed properly. The manuscript can be accepted after taking care of some minor issues.

  1. Still some moderate changes in English language is required throughout the manuscript. One example: line 54 "Active treatment adding alkaline chemicals such as..." this sentence is not appropriate. 
  2. Also, the manuscript is mixed of present and past tenses. While reporting any observation or analytical outcome you need to use past tense. For example, section "5.3" should be written in past tense instead of present tense. The authors should be revising the manuscript thoroughly.
  3. In the Methods section while describing the process of field data collection and analysis, mention the number of replicates were collected for each measurement. The authors have provided SD in the "Results" section, so this information is needed here.
  4. In the "Results" section while describing the field data output such as AMD flow rate use the SD along with the amount. For an example it was mentioned that "In little rain days, AMD flow rate was calculated at 0.059 m3/min after backfilling, which was 95.1% of the flow rate before backfilling.", without the information on SD it couldn't be claimed definitively that this falls inside or outside of the error margin as the change is very nominal.
  5. In figure 7, I assume that the numbers on top of the bars are SD. The authors should mention that in the figure caption for the clarity.
  6. Same comments go for figure 9 regarding the labeling of the SDs. Also, what is in the parenthesis? Is it meant for "±"? 
  7. In line 210 it has been written as "..in this chapter." As this is a journal article the phrase chapter is not appropriate here.
  8. The conclusion starts with a active voice "We constructed....". Use passive voice for maintaining proper scientific expression.

Author Response

(The authors gave the same response as above.)
